# The Gasotransmitter Hydrogen Sulfide (H_2_S) Prevents Pathologic Calcification (PC) in Cartilage

**DOI:** 10.3390/antiox10091433

**Published:** 2021-09-08

**Authors:** Sonia Nasi, Driss Ehirchiou, Jessica Bertrand, Mariela Castelblanco, James Mitchell, Isao Ishii, Alexander So, Nathalie Busso

**Affiliations:** 1Service of Rheumatology, Department of Musculoskeletal Medicine, Centre Hospitalier Universitaire Vaudois, University of Lausanne, 1011 Lausanne, Switzerland; drehirchiou@gmail.com (D.E.); Mariela.CastelblancoCastelblanco@unil.ch (M.C.); alexanderkai-lik.so@chuv.ch (A.S.); nathalie.busso@chuv.ch (N.B.); 2Department of Orthopaedic Surgery, Otto-von-Guericke University, 39120 Magdeburg, Germany; jessica.bertrand@med.ovgu.de; 3Department of Health Sciences and Technology, ETH Zürich, 8092 Zürich, Switzerland; 4Department of Health Chemistry, Showa Pharmaceutical University, Tokyo 194-8543, Japan; i-ishii@ac.shoyaku.ac.jp

**Keywords:** hydrogen sulphide, pathologic calcification, osteoarthritis, cartilage, animal models

## Abstract

Pathologic calcification (PC) is a painful and disabling condition whereby calcium-containing crystals deposit in tissues that do not physiologically calcify: cartilage, tendons, muscle, vessels and skin. In cartilage, compression and inflammation triggered by PC leads to cartilage degradation typical of osteoarthritis (OA). The PC process is poorly understood and treatments able to target the underlying mechanisms of the disease are lacking. Here we show a crucial role of the gasotransmitter hydrogen sulfide (H_2_S) and, in particular, of the H_2_S-producing enzyme cystathionine γ-lyase (CSE), in regulating PC in cartilage. Cse deficiency (Cse KO mice) exacerbated calcification in both surgically-induced (menisectomy) and spontaneous (aging) murine models of cartilage PC, and augmented PC was closely associated with cartilage degradation (OA). On the contrary, Cse overexpression (Cse tg mice) protected from these features. In vitro, Cse KO chondrocytes showed increased calcification, potentially via enhanced alkaline phosphatase (Alpl) expression and activity and increased IL-6 production. The opposite results were obtained in Cse tg chondrocytes. In cartilage samples from patients with OA, CSE expression inversely correlated with the degree of tissue calcification and disease severity. Increased cartilage degradation in murine and human tissues lacking or expressing low CSE levels may be accounted for by dysregulated catabolism. We found higher levels of matrix-degrading metalloproteases *Mmp-3* and *-13* in Cse KO chondrocytes, whereas the opposite results were obtained in Cse tg cells. Finally, by high-throughput screening, we identified a novel small molecule CSE positive allosteric modulator (PAM), and demonstrated that it was able to increase cellular H_2_S production, and decrease murine and human chondrocyte calcification and IL-6 secretion. Together, these data implicate impaired CSE-dependent H_2_S production by chondrocytes in the etiology of cartilage PC and worsening of secondary outcomes (OA). In this context, enhancing CSE expression and/or activity in chondrocytes could represent a potential strategy to inhibit PC.

## 1. Introduction

Pathologic calcification (PC) is a process whereby calcium-containing crystals deposit in soft tissues that normally do not calcify, such as cartilage, tendon, blood vessels, muscle and skin. PC exists in acquired and genetic forms, the latter being very rare [1]. Acquired PC affects up to 70% of patients who suffer musculoskeletal injuries (e.g., trauma or surgery to meniscus or tendons) [1,2]. In addition to pain and limitation of movement, PC can cause complications due to mechanical compression and inflammation of adjacent tissues. In cartilage, PC is an etiologic factor in the induction of cartilage matrix breakdown and osteoarthritis (OA) [3,4,5,6], and is found in 100% of cartilage samples from OA patients at the time of joint replacement [7].

Two families of calcium-containing crystals have been characterized in cartilage PC: basic calcium phosphate (BCP) crystals, of which the most common component is hydroxyapatite (HA), and calcium pyrophosphate dihydrate (CPPD) crystals [8]. In vitro, both crystals induce inflammatory responses, such as increased secretion of the pro-inflammatory cytokines IL-1β, IL-6, TNF-α and of prostaglandin E_2_ (PGE_2_) [4,9,10,11]. Moreover, they induce catabolic responses via induction of matrix metalloproteinases (MMP-1, -3, -9 and -13) [12] and suppression of their inhibitors (TIMP-1 and -3) [13,14,15]. CPPD crystals also trigger inflammatory cell death called necroptosis [16,17], whereas BCP crystals induce apoptosis, via Annexin V upregulation [18]. Finally, crystals induce oxidative stress via generation of reactive oxygen species (ROS) [19,20]. All these pathways can, in turn, amplify calcification in cells, as inflammatory cytokines [21], cell death [22] and ROS [23,24] have all been demonstrated to be pro-calcifying stimuli. Thus, understanding the mechanisms of crystal formation and deposition may be of therapeutic relevance in the context of cartilage PC seen in OA. 

Current therapies for PC, such as nonsteroidal anti-inflammatory drugs (NSAIDs) and local radiotherapy, are only symptomatic and do not block the progression of PC. Consequently, understanding the underlying mechanisms of PC is of paramount importance to identify new therapeutic targets to prevent or treat this condition.

Hydrogen sulfide (H_2_S) is a gasotransmitter [25] endogenously produced in mammalian cells by three enzymes: cystathionine beta-synthase (CBS), cystathionine gamma-lyase (CSE) and 3-mercaptopyruvate sulfurtransferase (3-MST), each of which has a unique tissue-specific pattern of expression. H_2_S has pleiotropic biological effects [26] relevant to PC [27]. We recently demonstrated that 3-MST deficiency exacerbated chondrocyte calcification in vitro and joint PC in vivo [24]. In the same models, administration of sodium thiosulfate (STS), an H_2_S donor, attenuated calcification [20]. However, the role of the other H_2_S-producing enzymes, CSE and CBS, is unclear. CSE activity has been linked to calcification of vascular tissues [28], and silencing of CSE and CBS increased human aortic valve calcification [29]. In contrast, CSE-generated H_2_S promoted bone healing [30] and two Cbs-deficient mouse models showed an osteoporotic phenotype [31]. 

In light of the above data, in the current study we examined the role of CSE in cartilage PC and investigated whether upregulating CSE activity can inhibit PC and OA progression. 

## 2. Materials and Methods

### 2.1. Mice

Female Cse knock-out (Cse KO) [32] and Cse transgenic (Cse tg) mice [33], both in the C57BL/6 background, and Cbs knock-out (Cbs KO) mice [34], in the C3H/HeJ background, were used. Control wild type (WT) mice, age-, sex- and background-matched, were used. For each experiment*,* 5 to 8 mice per group were used.

### 2.2. Murine Models of Cartilage PC

Spontaneous cartilage PC was studied in knees of aged (61–64 weeks old) WT and Cse KO mice. Surgically-induced cartilage PC was studied in 12 weeks old WT, Cse KO and Cse tg mice subjected to knee meniscectomy and sacrificed after 2 months [35]. Briefly, the medial meniscus was removed in the right knee, and the contralateral knee was sham-operated and used as control. For both experiments, knees were dissected and fixed in 10% formalin for 7 days for microCT-scan analysis.

### 2.3. Murine Knee microCT-Scan 

Knee images were acquired using a 1076 X-ray microCT-scan system (SkyScan, Bruker, MA, USA) and the following parameters: 18 μm resolution, 60 kV, 167 μA, 0.4° rotation step over 360°, 0.5 mm aluminum filter, 1180 min exposure time. The average scanning time was 25 min. Three-dimensional reconstruction was performed using NRecon V.1.6.6.0 (SkyScan, Bruker, MA, USA) and the following parameters: grey values = 0.0000–0.105867, ring artefact reduction = 3, beam hardening correction = 40%. Quantitative analyses of knee PC, in particular PC volume (mm^3^) and PC crystal content (µg), were conducted using CTAnalyzer V.1.10 (SkyScan, Belgium).

### 2.4. Murine Knee Histology 

After microCT-scan analysis, knees were decalcified in 10% formic acid for 10 days, dehydrated and paraffin embedded. Sagittal knee sections (6 µm) were stained with Safranin-O and counterstained with fast green/iron hematoxylin. For the surgically-induced PC model, 3 sections per mouse, spaced 70 µm apart, were scored through the medial knee compartment. For the spontaneous PC model, 3 sections per mouse, spaced 70 µm apart, were scored through the whole knee joint. For each mouse, the average score of the 3 sections/mouse was plotted in graphs. Histological scorings were assessed using the OARSI score [36], by two independent observers blinded with regard to the mice groups. Briefly, both OARSI scores (cartilage damage and Safranin-O loss) were obtained by multiplying the scored grade (depth progression into cartilage) by the scored stage (horizontal extent of cartilage involvement). Six grades were determined: 0 = intact cartilage surface; 1 = uneven cartilage surface; 2 = fibrillated cartilage surface; 3 = fissured cartilage; 4 = erosion till deep cartilage; 5 = bone surface denudation; 6 = bone remodeling and deformation. Four stages were also distinguished: 0 = no joint involvement; 1 ≤ 10% joint involvement; 2 = 10–25% involvement; 3 = 25–50% involvement; 4 ≥ 50% involvement. 

### 2.5. Murine Chondrocyte Isolation and Induction of Calcification 

Chondrocytes were isolated from 6 day old mice (Cse KO, Cse tg, Cbs KO and corresponding WT mice) as described previously [4]. Chondrocytes were amplified in DMEM with 10% FBS and 1% Penicillin Streptomycin and used at P1. For crystal formation analysis, cells were cultured for 14 days in calcification medium (BGJb, Thermo Fisher Scientific, Waltham, MA, USA) medium with 10% FBS, 0.2 mM L-ascorbic acid, 2-phosphate, 20 mM β-glycerol phosphate and 1% Penicillin Streptomycin). Medium and stimulations were refreshed each 3 days of culture. Alternatively, in some experiments, cells were cultured for 24 h in medium supplemented with secondary calciprotein particles (CPP, final concentration equivalent to 100 mg/mL calcium) to induce calcification, as previously described [37]. Cells were treated with sodium thiosulfate (STS 25 mM, Sigma-Aldrich, St. Louis, MI, USA), homocysteine (Hcy 25 or 100 µM, Sigma) or the CSE PAM (50 µM, provided by Exquiron Z401688062 = EXQ0160324, purity 93%, here named SU004) where indicated.

### 2.6. Quantification of Chondrocyte Calcification

After incubation in calcification media, cell supernatants were collected for ELISA and LDH measurement. Cell monolayers were washed in PBS, fixed in 10% formol and subjected to crystal deposition analysis through Alizarin red staining and quantified, as previously described [38]. For some experiments, quantification of calcium content in the cell monolayer was performed by the QuantiChrom^TM^ Calcium Assay Kit (BioAssay Systems, Hayward, California, United States) as specified by the manufacturer’s protocols, and absorbance was read at 612 nm using a Spectramax M5e plate reader (Molecular Devices, San Jose, CA, USA).

### 2.7. Stimulation with Calcium Phosphate Crystals 

Hydroxyapatite (HA) crystals were synthesized and characterized as previously described [39]. HA crystals were sterilized by gamma-radiation and pyrogen-free (≤0.01 EU/10 mg by Limulus amebocyte cell lysate assay). Prior to experimentation, crystals were resuspended in sterile PBS and sonicated for 5 min. 

### 2.8. Alkaline Phosphatase Activity

Chondrocyte supernatants were removed and alkaline phosphatase (Alpl) activity was measured in cell lysate using a p-Nitrophenyl Phosphate assay (Alkaline Phosphatase Assay Kit, ab83369, Abcam, Cambridge, UK). Absorbance was read at 405 nm using the Spectramax M5e plate reader, accordingly to the manufacturer’s protocol.

### 2.9. Real Time PCR Analysis

RNA was extracted (RNA Clean & Concentrator5, Zymoresearch, Irvine, CA, USA) and reverse transcribed (Superscript II, Invitrogen, Waltham, Massachusetts, United States), and quantitative Real Time PCR (qRT-PCR) analysis with gene specific primers using the LightCycler480^®^system (Roche Applied Science, Penzberg, Germany) was performed (Table 1). Murine data were normalized against *Tbp* and *Gapdh* reference genes, with fold induction of transcripts calculated against control cells. 

### 2.10. Human Cartilage Samples

Human OA articular cartilage was obtained from 15 patients undergoing knee joint replacement (Kellgren–Lawrence score from 1 to 4, 64.69 ± 10.58 years) from the Otto-von-Guericke University (Magdeburg). Full thickness cartilage pieces were resected from the main loading area of the medial compartment of both tibia and femur. Cartilage tissue was fixed in freshly prepared 4% paraformaldehyde for immunohistochemical and calcification analysis.

For the HA crystal stimulation experiment, cartilage from the tibial plateau and femoral condyles from 4 OA patients (mean age 72 ± 10 years) undergoing total knee replacement (Kellgren and Laurence (KL) score = 4), obtained from the Orthopedics Department (DAL, CHUV, Lausanne-CH), was used. Six-millimeter diameter disks (3 disks/patient) were dissected from macroscopically intact cartilage using a dermal punch. To match for location across treatment groups, each disk was divided in two equal parts, and each half was stimulated or not with 500 µg/mL of HA crystals for 24 h in individual 96 wells, coated with Poly (2-hydroxyethyl methacrylate) in culture medium (DMEM + 1% Penicillin Streptomycin + 50 µg/mL L-ascorbic acid 2-phosphate). After the incubation period, cartilage was recovered for CSE immunohistochemistry.

For chondrocyte isolation, cartilage from tibial plateau and femoral condyles from 3 OA patients (mean age 70 ± 8 years) undergoing total knee replacement (Kellgren and Laurence (KL) score = 4), obtained from the Orthopedics Department (DAL, CHUV, Lausanne-CH), was used.

### 2.11. Human Chondrocyte Isolation and Induction of Calcification 

Chondrocytes were isolated from cartilage pieces incubated overnight in the digestion enzyme Liberase TM (Roche). For chondrocyte crystal formation analysis, cells were cultured for 24 h in medium with secondary calciprotein particles (CPP, working concentration equivalent to 100 mg/mL calcium) to induce calcification, as described previously [37].

### 2.12. Human Cartilage Histology and Quantification of Calcification

Paraffin sections (4 μm) were cut and stained with von Kossa/Safranin-Orange staining (Sigma). OARSI scoring (from 0 to 5) was performed and cartilage samples were grouped into low (K/L 1–2 and OARSI 2–3), medium (K/L 3 and OARSI 3–4), and high (K/L 4 and OARSI 5) OA grades. Quantitative measurement of the areas of total cartilage mineralization was performed using ImageJ (NIH Image [40]) by setting the upper and lower bounds of the threshold utility to a specified level for all samples. These areas were used for calculating the percentage of calcification over the whole cartilage area. 

### 2.13. Immunohistochemical Analysis 

Human CSE expression was evaluated using an anti-CSE rabbit polyclonal antibody (Sigma, HPA-023300). Murine Cse expression was evaluated using an anti-Cse rabbit polyclonal antibody (Proteintech, CSE 12217-1-AP). Human and murine CBS expression was evaluated using an anti-CBS goat polyclonal antibody (SC46830, Santa Cruz Biotechnology, Dallas, TX, USA). Apoptotic chondrocytes were detected using the ApopTag plus Peroxidase In situ Kit (MilliporeSigma, Burlington, MA, USA). as previously described [41]. All stainings were performed on paraffin sections. The percentage of positive cells over the total number of cells was quantified in three different fields for each sample and the mean plotted in the graphs. 

### 2.14. H_2_S Measurements 

Chondrocytes from Cse KO, Cse tg, Cbs KO and corresponding WT mice were resuspended in fluorescence-activated cell sorting (FACS) buffer (5% FCS, 5 mM EDTA in PBS). For each condition, 10^6^ cells were incubated with the H_2_S fluorescent probe P3 (10 µM) [42] and after two minutes FACS analysis was performed on a LSRII SORP cytometer (BD Biosciences, Franklin Lakes, NJ, USA) with a UV laser. FACS Diva (BD Biosciences) and FlowJoX (Tree Star, Ashland, OR, USA) software were used for data processing. 

### 2.15. IL-6 Quantification

Cell supernatants were collected at the reported time-points and assayed using a murine or human IL-6 ELISA kit (eBioscience). The manufacturer’s protocols were explicitly followed, and absorbance was read at 45 nm and 570 nm using the Spectramax M5e plate reader.

### 2.16. LDH Measurement

LDH in supernatant was measured using CytoTox-ONE™ Homogeneous Membrane Integrity Assay (Promega, Madison, WI, USA) according to the manufacturer’s instructions. LDH release (%) was calculated using the following formula: LDH release (%) = [(value in sample) − (background)]/[(value in Triton X100 treated sample) − (background)] × 100.

### 2.17. Statistical Analysis 

For in vitro experiments, values represent means ± SD of triplicates from one representative experiment of three independent experiments. For in vivo experiments, 5 to 8 mice per group were used. Data were analyzed with GraphPad Prism software (GraphPad software), San Diego, CA. Variation between data sets was evaluated using the Student’s *t* test or one-way or two-way ANOVA test, where appropriate. Correlations were evaluated with Pearson’s *r* correlation. Differences and correlation were considered statistically significant at * *p* < 0.05, ** *p* < 0.01, *** *p* < 0.001, **** *p* < 0.0001.

## 3. Results

### 3.1. Cse Expression Regulates Pathologic Calcification and Cartilage Damage in Murine Knees

The role of Cse in PC after knee meniscectomy (MNX) was studied. Chondrocytes were stained for Cse in the cartilage of sham-operated mice while Cse expression was significantly reduced in cartilage from MNX mice (Figure 1a). The effect of Cse expression on PC and cartilage damage was investigated in Cse KO and Cse tg mice subjected to MNX, and compared with their respective WT mice. Cse KO mice had increased intra-articular PC compared to WT at 2 months post-surgery (Figure 1b, white arrows). Quantitative analysis confirmed increased PC volume and PC crystal content in Cse KO mice (Figure 1c). 

Histological features of OA were also quantified. Cse KO joints showed exacerbated cartilage damage and proteoglycan loss (Safranin-O staining) compared to WT (Figure 1d). This was reflected in the significantly increased OARSI scores in Cse KO mice (Figure 1e). 

The role of Cse in PC in spontaneous aging was also studied. Aged Cse KO mice (61–64 weeks) displayed larger peri-articular calcific deposits compared to WT littermates (Appendix A, white arrows), although quantitative measurements did not reach statistical significance (Appendix A). Aged Cse KO mice showed significantly higher cartilage degradation and features of OA (Appendix A).

The effect of increased Cse was studied using Cse tg mice. They showed less severe MNX-induced calcification (Figure 1f,g). Concomitantly, cartilage damage was significantly reduced (Figure 1h,i). Finally, Cse deficiency or overexpression did not significantly alter the number of apoptotic chondrocytes (Appendix A). 

We then assessed the effect of Cse on cartilage metabolism, in particular the transcription of matrix-degrading enzyme genes (*Mmp-3* and *Mmp-13*) and their inhibitors (*Timp-1* and *Timp-3*) in WT, Cse KO and Cse tg chondrocytes in vitro. Cse KO cells showed significantly increased expression of *Mmp-3* and *Mmp-13*, and decreased *Timp-1* expression relative to WT cells (Figure 1l). On the contrary, *Mmp-3* and *Mmp-13* were decreased and *Timp-3* strongly increased in Cse tg chondrocytes. 

Overall, these results demonstrated that a catabolic chondrocyte gene expression signature was exacerbated by Cse deficiency and attenuated by Cse overexpression, and that Cse expression protects against joint calcification and cartilage damage. 

### 3.2. Cse Regulates Chondrocyte Calcification and IL-6 Secretion In Vitro

The link between chondrocyte calcification and H_2_S levels was examined in vitro. Chondrocyte calcification started after 7 days of culture and increased over time (Figure 2a), whereas H_2_S levels were reduced by approximately 40% after 7 days and by 60% after 10 days (Figure 2b). 

We then studied the specific effect of Cse modulation in this context. First, we determined that WT chondrocytes expressed *Cse* in addition to genes for the other H_2_S-producing enzymes *3-Mst* and *Cbs* (C_t_ *Cse* = 28, C_t_ *3-Mst* = 26, C_t_ *Cbs* = 30). Undetectable *Cse* expression in Cse KO chondrocytes was associated with decreased *3-Mst* expression, whereas *Cbs* expression was similar to that in WT chondrocytes (Figure 2c). Conversely, increased *Cse* expression in Cse tg chondrocytes was associated with a seven-fold upregulation of *3-Mst* expression, whereas *Cbs* expression was similar to that of WT chondrocytes (Figure 2c). Taken together, these results suggest that the lack of *Cse* expression is not compensated by *3-Mst* or *Cbs.* Indeed, Cse KO chondrocytes produced significantly less H_2_S, while Cse tg chondrocytes produced significantly more H_2_S, compared to WT chondrocytes (Figure 2d).

In calcification assay, Cse KO chondrocytes calcified significantly more compared to WT cells (Figure 2e), whereas Cse tg chondrocytes showed significantly decreased calcification (Figure 2g), as demonstrated by Alizarin red staining and by quantification of calcium content. Concomitantly, we observed that IL-6, a pro-calcification cytokine [4], was significantly increased in Cse KO chondrocytes (Figure 2f) and decreased in Cse tg cells (Figure 2h). 

The activity of the calcification-enzyme alkaline phosphatase (Alpl) was assayed in chondrocytes with different Cse genotypes. Basal Alpl activity in Cse KO chondrocytes was significantly increased relative to WT cells, whereas Cse tg cells showed no difference (Figure 2i), suggesting that the hyper-calcification phenotype seen in Cse KO chondrocytes may be due in part to increased Alpl activity.

As chondrocyte hypertrophy is known to be associated with calcification, we compared gene expression of differentiation markers. Cse KO chondrocytes preferentially expressed hypertrophic-related genes such as *Coll10* and *Runx2*, rather than early-stage differentiation genes such as *Coll2* and *Sox9*. This was determined by qRT-PCR (ratio *Coll10/Coll2* = 3 in *Cse* KO cells, ratio = 1 in WT cells; ratio *Runx2/Sox9* = 1 in *Cse* KO cells, ratio = 0.6 in WT cells).

Increased levels of homocysteine have been reported in Cse-deficient mice and cells [32], and may induce pro-calcification mechanisms such as mitochondrial dysfunction, apoptosis and oxidative stress [43]. Therefore, we wanted to rule out that the pro-calcification phenotype seen in Cse-deficient cells and mice could be accounted for by increased homocysteine levels. We incubated WT chondrocytes with different concentrations of homocysteine (100 or 25 µM) and found significantly decreased calcification (Figure 2l), therefore precluding a role of hyperhomocysteinemia in the exacerbated calcification phenotype of Cse KO.

### 3.3. CSE Expression in Human Cartilage Negatively Correlates with Tissue Calcification and Disease Severity

We explored the relationship between CSE expression and PC in humans. In cartilage from 14 OA patients undergoing knee joint replacement, *CSE* gene expression was found in cartilage from the macroscopically damaged area and was decreased compared to the undamaged cartilage area (data not shown). By immunohistochemistry, CSE was expressed in chondrocytes of the superficial and tangential cartilage layers, whereas little expression was seen in deep cartilage and close to the bone (Figure 3a). Contrarily, von Kossa staining for calcification (black spots) was mostly positive in deep cartilage areas (Figure 3a); thus, in OA cartilage, CSE expression and calcification appear to be mutually exclusive. We found a significant and inverse (*p* = 0.03, r = −0.57) correlation between CSE expression and chondrocyte calcification (Figure 3a, graph). When specimens from OA patients were subdivided by severity (K/L from 1 to 4) and OARSI (0–5) scores into low OA (K/L 1–2 and OARSI 2–3; 70.25 ± 5.37 years), medium OA (K/L 3 and OARSI 3–4; 65.25 ± 14.88 years) and high OA (K/L 4 and OARSI 5; 59.8 ± 9.33 years), CSE expression decreased by 30% in medium and high OA (Figure 3b, upper panel), whereas the extent of cartilage calcification increased in more severe OA (Figure 3b, lower panel). 

To understand the negative correlation between CSE expression and cartilage calcification, we macroscopically incubated non-eroded cartilage explants from four patients with 500 µg/mL of HA crystals for 24 h. CSE expression significantly decreased following incubation with crystals (Figure 3c). Overall, these results indicate that cartilage calcification increased during OA progression and the presence of HA crystals further impacted CSE expression. 

### 3.4. A CSE Positive Allosteric Activator Reduces Chondrocyte Calcification and IL-6 Secretion 

Our results demonstrated an inverse correlation between CSE expression in chondrocyte or tenocytes and calcification, IL-6 secretion and cartilage matrix catabolism (Figure 1, Figure 2 and Figure 3). We therefore hypothesized that a CSE allosteric activator that increases cellular H_2_S may counter these effects and be of therapeutic value in musculoskeletal calcification. We developed a screening assay using human recombinant CSE (hCSE) and L-Cysteine as substrate, and as a read-out we used the production of pyruvate as a surrogate of H_2_S production (Appendix A). Pyruvate was detected only in the presence of hCSE (Appendix A), increased in the presence of the CSE activator calmodulin [44], and almost completely inhibited by AVG, a known CSE inhibitor [45] (Appendix A). Based on these results, we screened 75,000 compounds from a lead-like small molecule library (steps in Appendix A). We identified one compound, named SU004 (Appendix A), which increased pyruvate in a dose-dependent manner (Appendix A) with an estimated EC50 of 17 µM. 

In chondrocyte calcification assay, this compound potently inhibited calcification (Figure 4a), with no cell toxicity (Figure 4b). The specificity of SU004 for CSE was confirmed using Cse-deficient chondrocytes, as Cse KO cells showed no change in calcification (Figure 4a). Finally, SU004 also blocked IL-6 secretion to the same extent as the positive control STS (Figure 4c). We finally confirmed SU004′s activity in human primary chondrocytes. In cells cultured from three independent patients (P1, P2, P3) we observed decreased calcification (Figure 4d) and decreased IL-6 secretion (Figure 4f) with no effect on cell cytotoxicity (Figure 4e).

## 4. Discussion 

The mechanisms leading to pathologic calcification in musculoskeletal tissues are poorly understood and no medical therapy currently exists, with the exception of surgery. Our data uncovered a role for the H_2_S-producing enzyme CSE in cartilage PC (Figure 5). Indeed, increased cartilage calcification is seen when Cse is absent, in surgically-induced and aging murine models. Pathologically, we also found that the amount of calcification and the histological severity of OA in mice and humans were negatively correlated with CSE expression. The in vitro results showed that Cse deficiency led to reduced cellular levels of H_2_S and increased calcification in chondrocytes. When H_2_S levels were enhanced pharmacologically in chondrocytes using a new small-molecule CSE PAM, or genetically by Cse overexpression, reduced calcification was observed. These studies indicate that CSE-generated H_2_S is a regulator of experimental and human cartilage PC. 

We previously showed that administration of STS, which is transformed to H_2_S, reduced the severity of experimental OA. There are multiple sources of endogenous H_2_S in joint tissues. We recently found that 3-MST is abundantly expressed in chondrocytes and its deficiency exacerbated PC in vitro and in vivo [24]. The third H_2_S-producing enzyme CBS is expressed at low levels in human cartilage (Appendix A) and is downregulated in osteoarthritic mice (Appendix A). The impact of Cbs in in vivo models cannot be studied, because less than 25% of Cbs KO mice survive to adult age [34]. Cbs deficiency, similarly to Cse deficiency, led to increased chondrocyte calcification and increased IL-6 production (Appendix A), although with a milder phenotype than that observed in Cse KO chondrocytes. This can be explained by compensatory mechanisms of Cse and 3-Mst in the absence of Cbs, as evidenced both at the gene expression level (Appendix A) and by unchanged H_2_S production in Cbs KO chondrocytes (Appendix A). In conclusion, all three H_2_S producing enzymes can modulate calcification, but this is the first time that a role for CSE has been proven.

Because CSE deficiency leads to the accumulation of its substrate homocysteine [32], a known factor of increased vascular calcification in humans [46,47], the pro-calcifying phenotype seen in Cse KO may be due to homocysteine accumulation. Our results ruled out its contribution to calcification because raised homocysteine levels decreased chondrocyte calcification. 

Other data support a protective role for H_2_S in vascular calcification in rats [48,49] and calcification of human VSMCs in vitro [28,37,50]. These findings, however, contrast with reports of H_2_S as a promotor of physiological calcification [51,52,53,54] and severe skeletal abnormalities in Cbs-deficient mice [31,55] and in patients with CBS deficiency [56]. However, in both mice [32] and humans [57] deficient for CSE, there is no evidence of skeletal abnormalities, suggesting that CSE is involved in pathological but not physiological calcification.

A major mechanism of H_2_S’s cellular effects could be via post-translational modification such as protein persulfidation. H_2_S can modify cysteine thiols or disulfide bonds, thereby changing protein function [58]. Relevant proteins include NF-κB, whose anti-apoptotic transcriptional activity was increased upon persulfidation [59]. Similarly, persulfidation of Keap-1 [60,61] stabilized Nrf2 and led to activation of genes including *HO-1* and *NQO1* [37], and inhibited osteoblast [62] and VSMC mineralization [37], respectively. Finally Alpl contains disulfide bonds that are critical for its function [63]. In this respect, we found that Cse KO chondrocytes and tenocytes both calcified more and showed increased basal levels of Alpl expression or activity. Definitive proof will require the demonstration of persulfidation of the Alpl protein. Finally, H_2_S may exert an anti-calcifying effect via inhibition of IL-6 signaling [20,24,64,65,66]. Therefore, CSE-generated H_2_S can block the previously proven vicious cycle between calcification and inflammation [4]. Finally, tissue CSE regulates the balance of cartilage extracellular matrix degradation via H_2_S production.

Based on the above findings, we questioned if enhancing CSE-mediated H_2_S production may be clinically relevant. The H_2_S donor sodium thiosulfate (STS) prevented cartilage calcification and was chondroprotective in mouse OA [20]. Because long-term administration of STS is not clinically feasible, we searched for an allosteric modulator that can increase H_2_S production by acting on CSE. Our results indicate that this approach is feasible, and the molecule we identified, namely SU004 (6′,7′-dimethoxy-3′-methyl-3,3′,4,4′-tetrahydro-1H,1′H-spiro[naphthalene-2, 2′-quinazoline]-4′-one) is able to inhibit calcification and IL-6 secretion in murine and human chondrocytes, similar to STS. This suggests the potential for the development of a novel therapeutic class of H_2_S modulators, based on augmenting CSE activity that can be tested clinically.

## 5. Conclusions

In conclusion, through in vitro, in vivo, and human data, we established a key role for CSE-generated H_2_S in the regulation of pathological chondrocyte calcification. This pathway is a potential target for the development of disease-modifying drugs for osteoarthritis and other pathologic conditions associated with calcification.

## Figures and Tables

**Figure 1 antioxidants-10-01433-f001:**
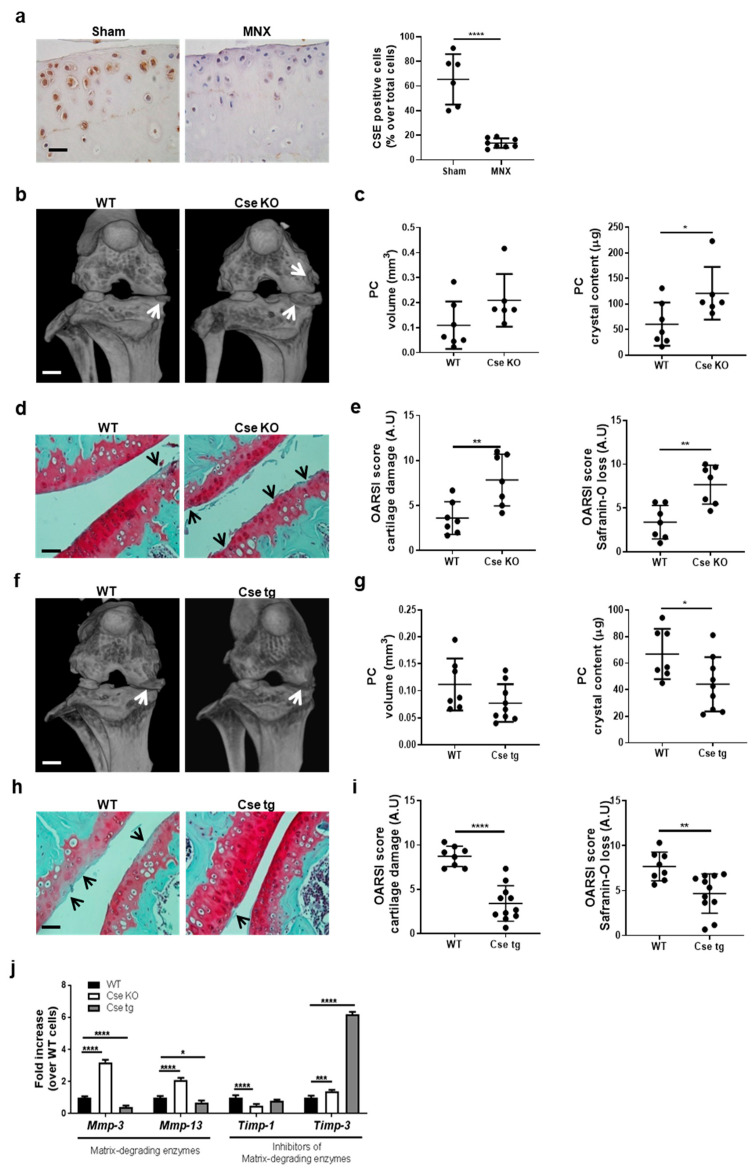
Cse deficiency exacerbates joint calcification and cartilage damage in an experimental model of knee PC. (**a**) Representative immunohistochemical analysis of Cse expression in knee sections from sham-operated and MNX WT mice. A knee section from a sham-operated Cse KO mouse was used to prove the specificity of the Cse antibody (data not shown). Scale bars 150 µm. (**b**) Representative micro-CT scan images of WT and Cse KO murine knee joints two months after meniscectomy (MNX). White arrows show calcification in MNX WT knees and its exacerbation in Cse KO mice. Scale bars 1 mm. (**c**) Quantitative analysis of PC volume (mm^3^) and PC crystal content (µg) in WT and Cse KO MNX knees. (**d**) Representative histologies of WT and Cse KO MNX knees, stained with Safranin-O. Black arrows show increased degenerative OA changes in articular cartilage of Cse KO mice. Scale bars 150 µm. (**e**) Scoring of cartilage damage and Safranin-O loss, according to the OARSI method. Mice number WT n = 7, Cse KO n = 7. (**f**) Representative micro-CT scan images of WT and Cse tg MNX murine knee joints two months after surgery. White lines show pathologic calcification in MNX WT knees and its decrease in Cse tg mice. Scale bars 1 mm. (**g**) Quantitative analysis of PC volume (mm^3^) and PC crystal content (µg) in WT and Cse tg MNX knees. (**h**) Representative histologies of WT and Cse tg MNX knees, stained with Safranin-O. Black arrows show decreased degenerative OA changes in articular cartilage of Cse tg mice. Scale bars 150 µm. (**i**) Scoring of cartilage damage and Safranin-O loss, according to the OARSI method. Mice number WT n = 7–8, Cse tg n = 9–11. (**j**) qRT-PCR analysis of the indicated genes in WT, Cse KO and Cse tg chondrocytes, at basal level. WT cells are considered as reference (=1). Statistics: * *p* < 0.05, ** *p* < 0.01, *** *p* < 0.001, **** *p* < 0.0001.

**Figure 2 antioxidants-10-01433-f002:**
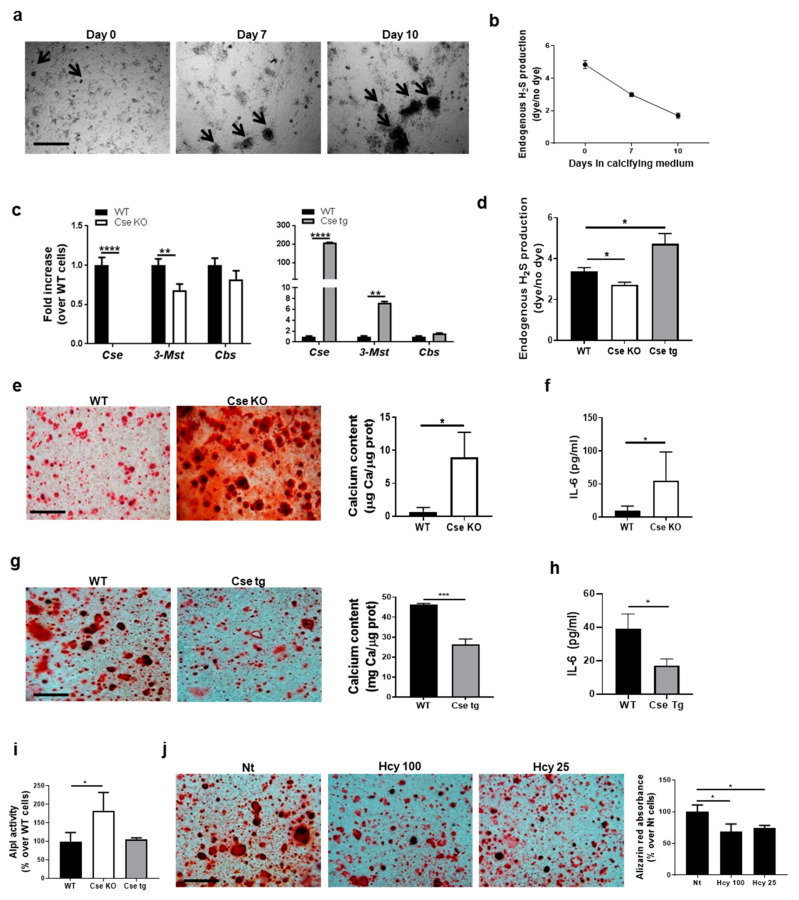
Cse-generated H_2_S regulates calcification and IL-6 secretion in chondrocytes. (**a**) Chondrocytes cultured in calcifying medium for 0, 7 and 10 days. Black arrows indicate calcium-containing crystals. Scale bars 500 µm. (**b**) FACS analysis by P3 probe of endogenous H_2_S production by chondrocytes in (**a**). (**c**) qRT-PCR analysis of the indicated genes in WT, Cse KO and Cse tg chondrocytes, at basal level. WT cells are considered as reference (=1). (**d**) FACS analysis by P3 probe of endogenous H_2_S production by WT, Cse KO and Cse tg chondrocytes, at basal level. (**e**) Alizarin red staining of WT and Cse KO chondrocytes cultured in calcification medium for 14 days. Scale bars 500 µm. Graph represents calcium content in the cell monolayer, expressed in mg Ca/µg protein. (**f**) IL-6 secretion by WT and Cse KO chondrocytes in (**f**). (**g**) Alizarin red staining of WT and Cse tg chondrocytes cultured in calcification medium for 14 days. Scale bars 500 µm. Graph represents calcium content in the cell monolayer, expressed in mg Ca/µg protein. (**h**) IL-6 secretion by WT and Cse tg chondrocytes in (**h**). (**i**) Basal Alpl activity in WT, Cse KO and Cse tg chondrocytes, expressed as % over WT cells. (**j**) Alizarin red staining and quantification in WT chondrocytes cultured in calcification medium for 14 days and treated with different concentrations of homocysteine (Hcy 100 µM and Hcy 25 µM). Scale bars 500 µm. Graph represents alizarin red quantification in the cell monolayer, expressed in % over Nt cells. Statistics: * *p* < 0.05, ** *p* < 0.01, *** *p* < 0.001, **** *p* < 0.0001. Pictures represent triplicates from one experiment of three independent experiments. Graphs represent mean ± SD or triplicate samples from one experiment of three independent experiments.

**Figure 3 antioxidants-10-01433-f003:**
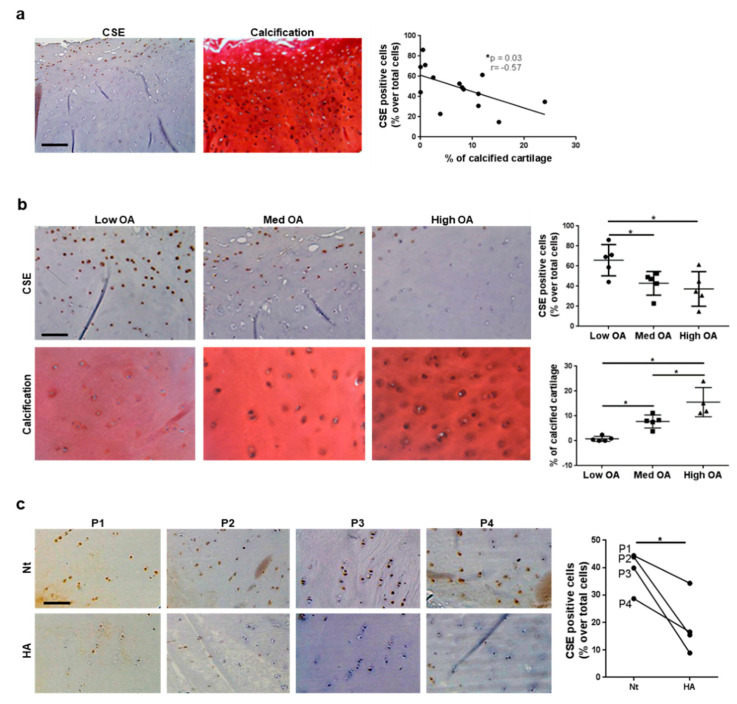
CSE expression negatively correlates with cartilage calcification in humans. (**a**) CSE immunohistochemical staining in cartilage from individuals with osteoarthritis (OA) and consecutive sections stained with von Kossa/Safranin−O staining for crystals. For each staining, one representative picture from one of 14 patients is shown. Scale bars 200 μm. The graph shows the correlation between the % of CSE positive cells and the % of calcified cartilage in the different patients. n = 14 patients. (**b**) CSE immunohistochemical staining in cartilage from individuals with low, medium and high stage osteoarthritis and consecutive sections stained with von Kossa/Safranin-O staining for crystals. For each staining, one representative picture from one of 5 patients per group is shown. Scale bars 200 μm. Graphs show quantification of the % of CSE positive cells and the % of calcified cartilage. For each patient, three fields were counted per explant and the mean plotted in the graph. n = 15 patients. (**c**) CSE immunohistochemical staining in human cartilage explants untreated (Nt) or stimulated with 500 µg/mL HA crystals for 24 h. Scale bars 200 µm. The graph shows the % of CSE positive cells in Nt and HA−stimulated explants in the different patients. For each patient, three fields were counted per explant and the mean plotted in the graph. Lines connect the Nt condition to the HA condition for each patient. n = 4 patients (P1−P4). Statistics: * *p* < 0.05.

**Figure 4 antioxidants-10-01433-f004:**
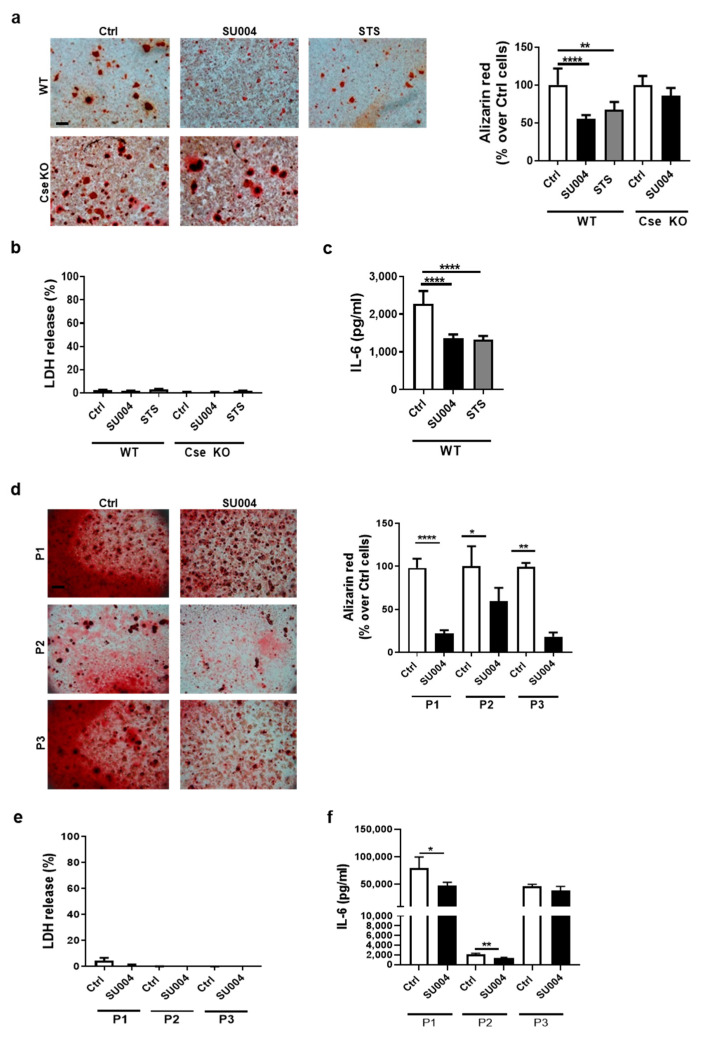
A new CSE allosteric activator inhibits chondrocyte PC and IL-6 production. (**a**) WT and Cse KO chondrocytes were incubated with the novel CSE activator (SU004, 50 µM) previously identified by high-throughput screening, or with the equivalent volume of DMSO (0.1%) for 24 h in calcification medium. STS 25 mM was used as a positive control for calcification inhibition. Calcium-containing crystals were stained with Alizarin red and quantified. Scale bars 200 µm. (**b**) Cytotoxicity was assessed by LDH measurement and expressed as % over 100% cytotoxicity. (**c**) IL-6 secretion was measured by ELISA in a supernatant of WT murine chondrocytes in (**a**). (**d**) Primary human chondrocytes from 3 OA patients (P1, P2, P3) were incubated with SU004 50 µM, or with the equivalent volume of DMSO (0.1%) for 24 h in calcification medium. Calcium-containing crystals were stained with Alizarin red and quantified. Scale bars 200 µm. (**e**) Cytotoxicity was assessed by LDH measurement and expressed as % over 100% cytotoxicity. (**f**) IL-6 secretion was measured by ELISA in supernatant of human chondrocytes in (**d**). Statistics: * *p* < 0.05, ** *p* < 0.01, **** *p* < 0.0001. Pictures represent triplicates from one experiment of three independent experiments. Graphs represent mean ± SD or triplicate samples from one experiment of three independent experiments.

**Figure 5 antioxidants-10-01433-f005:**
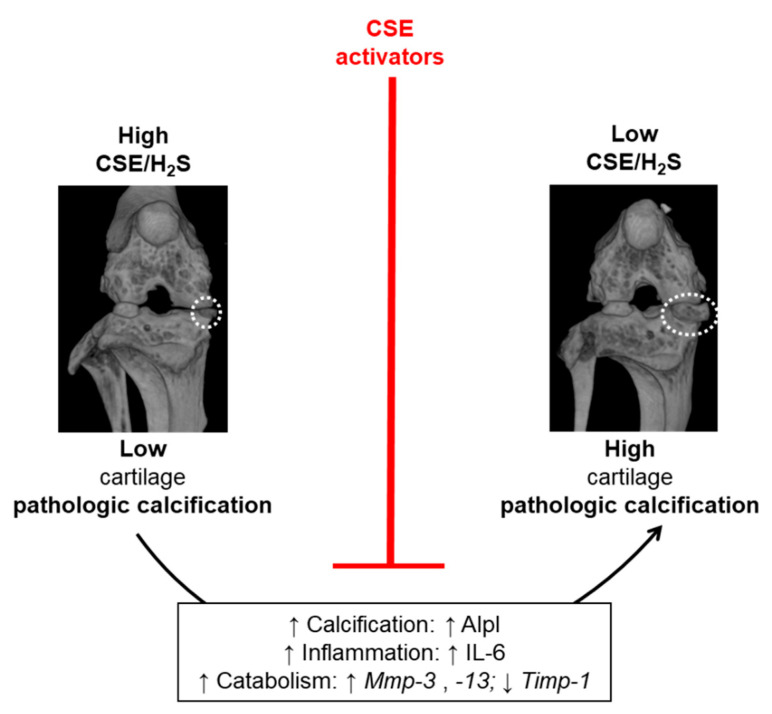
Proposed mechanism for CSE involvement in cartilage PC. High CSE expression, therefore high levels of H_2_S, in knee cartilage are associated with low pathologic calcification. Conversely, low CSE expression, and decreased H_2_S levels, favor pathologic calcification in cartilage. In particular, we described that lack of CSE sustains chondrocyte calcification in vitro and in knee in vivo, partly via increased Alp expression and/or activity. In addition, low CSE expression is associated with increased IL-6, which is known to amplify calcification, as previously described [4]. Finally, low CSE combines with increased Mmp-3 and Mmp-13 expression and decreased Timp-1 expression in vitro, leading to tissue degeneration. Restoring physiological levels of H_2_S in cartilage by CSE activators can block cell calcification and IL-6 production, and may therefore be of therapeutic relevance for the treatment of pathologic calcification in osteoarthritis.

**Table 1 antioxidants-10-01433-t001:** Murine gene specific primers for qRT-PCR.

Gene	Forward Primer (5′→3′)	Reverse Primer (5′→3′)
** *Cbs* **	AGC AAC CCT TTG GCA CAC TA	CTT ATC CAC CAC CGC CCT G
** *Cse* **	GCC AGT CCT CGG GTT TTG AA	TTG TGG TGT AAT CGC TGC CT
** *Coll2* **	ACA CTT TCC AAC CGC AGT CA	GGG AGG ACG GTT GGG TAT CA
** *Coll10* **	AAA CGC CCA CAG GCA TAA AG	CAA CCC TGG CTC TCC TTG G
** *Runx2* **	GGG AAC CAA GAA GGC ACA GA	TGG AGT GGA TGG ATG GGG AT
** *Sox9* **	AAG ACT CTG GGC AAG CTC TGG A	TTG TCC GTT CTT CAC CGA CTT CCT
** *Gapdh* **	CTC ATG ACC ACA GTC CAT GC	CAC ATT GGG GGT AGG AAC AC
** *Mmp-3* **	ATA CGA GGG CAC GAG GAG	AGA AGT AGA GAA ACC CAA ATG CT
** *Mmp-13* **	GCA GTT CCA AAG GCT ACA AC	GCT GGG TCA CAC TTC TCT G
** *3-Mst* **	CTG GGA AAC GGG GAG CG	GCT CGG AAA AGT TGC GGG
** *Tbp* **	CTT GAA ATC ATC CCT GCG AG	CGC TTT CAT TAA ATT CTT GAT GGT C
** *Timp-1* **	CCC ACA AGT CCC AGA ACC GCA G	GCA GGC AAG CAA AGT GAC GGC
** *Timp-3* **	TCC TAG ACC CAG TTC CAT ATA CAC TTC	TTG GAC TTC TGC CAA CTT CCT T

## Data Availability

Data is contained within the article or Appendix A.

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
