# Peer review of "The Gasotransmitter Hydrogen Sulfide (H2S) Prevents Pathologic Calcification (PC) in Cartilage"

_antioxidants, 2021, doi:10.3390/antiox10091433_

Round 1

Reviewer 1 Report

In my opinion this is a very good paper. Excellent presentation of results. Photos of calcification are evident for experimental groups. Nice and clear presentation of proposed mechanism for CSE involvement in cartilage PC. Good logical plan of experiments and therefore drawing conclusions from results is convincing. The results have high possible application in practice. I would only add the name of the identified molecule (PAM) in the sentence below pasted: Our results indicate that this approach is feasible, and the molecule we identified is able to inhibit calcification and IL-6 secretion in murine and human chondrocytes, similar to STS. lines 465-466. 

Author Response

We thank reviewer 1 very much for the positive feedback and her/his interest in our work.

As suggested, we added the name of the newly identified PAM in the text. The sentence is now: "Our results indicate that this approach is feasible, and the molecule we identified, namely SU004 (6’,7’-dimethoxy-3’-methyl-3,3’,4,4’-tetrahydro-1H,1’H-spiro[naphthalene-2, 2’-quinazoline]-4’-one) is able to inhibit calcification and IL-6 secretion in murine and human chondrocytes, similar to STS".

Reviewer 2 Report

The authors performed a study testing the hypothesis that upregulating CSE can inhibit pathological calficication  and osteoarthritis in mouse model and in smaples of human articular cartilage. Overall, the study is well justified and carefully performed. However, the following points should be addressed to make the paper more beneficial to the readers, namely:

2.1, lines 81-54 - the number of mice per study group is missing.

2.3, what was the average scanning time?

2.4., lines 101-109: What objective magnification was used? How many image fields were samples per section and animal? Also, briefly explain the parameters of the OARSI score and heir biological interpretation as this is fundamental for interpreting your data. Also correct the typos in micrometers consistently (e.g. 70 μM --> 70 μm).

2.10 Rename the section. "Human experiments" does not seem to be appriopriate to what you did. First, it was human articular cartilage, not human. Second, you describe an analysis rather than an experiment. 

2.12. How many sections? How many microscope image fields? Provide an image illustrating the quantification technique used in the ImageJ software.   Provide citation for the ImageJ according to their webpage.

You should definitely provide the complete primary data along with the manuscript, either as an electronic supplement or as a reference to any scientific database. 

The histological design is outdated in terms of the present tools of quantitative histology and stereology as one if its golden standards (see e.g. PMID 23769130). If you can not change the quantitative histological part, try at least discuss the pros and cons of your methodology when compared on papers using design-based stereological techniques when quantifying cartilage (e.g. PMID 32979538, 27562858, 32751860, 30036886 or ather papers).

Since section 3.2., there seems to be a formatting issue in the submitted file - there is a significant indentation on the left margin.

Minor typos should be corrected, missing spaces such as 100μM-->100 μM (line 305). This is on multiple places in the manuscript.

Supplementary figure 1 c is missing a scale bar, although it is mentioned in the figure legend.

Provide scale bars also for the micro-CT scan images.

Author Response

Please see point-by-point answers in the attachment
